# A programmable seekRNA guides target selection by IS*1111* and IS*110* type insertion sequences

Rezwan Siddiquee [1], Carol H. Pong [1], Ruth M. Hall [1] ✉ &
Sandro F. Ataide [1] ✉

IS*1111* and IS*110* insertion sequence (IS) family members encode an unusual DEDD transposase type and exhibit specific target site selection. The IS*1111* group include identifiable subterminal inverted repeats (sTIR) not found in the IS*110* type[1]. IS in both families include a noncoding region (NCR) of significant length and, as each individual IS or group of closely related IS selects a different site, we had previously proposed that an NCR-derived RNA was involved in target selection[2]. Here, we find that the NCR is usually downstream of the transposase gene in IS*1111* family IS and upstream in the IS*110* type. Four IS*1111* and one IS*110* family members that target different sequences are used to demonstrate that the NCR determines a short seeker RNA (seekRNA) that co-purified with the transposase. The seekRNA is essential for transposition of the IS or a cargo flanked by IS ends from and to the preferred target. Short sequences matching both top and bottom strands of the target are present in the seekRNA but their order in IS*1111* and IS*110* family IS is reversed. Reprogramming the seekRNA and donor flank to target a different site is demonstrated, indicating future biotechnological potential for these systems.

The majority of autonomous mobile elements found in bacteria that use a transposition mechanism, i.e. insertion sequences (IS) and transposons, exhibit little or no target site selectivity[3,4]. A few examples of highly selective target site and orientation specificity that involve additional proteins have been studied[5–9], and recently, a new family of IS that exhibits target site and orientation specificity using a single IS-encoded protein has been identified[10,11]. However, the flexible target selection seen in the IS*110* and IS*1111* families has not been investigated. IS in these families encode an unusual transposase type, referred to as DEDD transposases due to the presence of an N-terminal RuvC-like catalytic domain that includes the DEDD residues[12,13], but each member or group of closely related members exhibit specificity for a different site where they are found in only one orientation[4,14]. The founding IS of this type, IS*110*, was identified in the 1980s[15] and soon acquired relatives[3]. Later, IS related to IS*1111*[16], which encoded more distantly related transposases and had further distinguishing features,

were proposed to form a separate family[17]. However, to date, how the IS in these groups identify the appropriate target and move to it has received little attention[14,18].

A comprehensive search for IS related to IS*1111* was conducted 20 years ago, and a detailed analysis of over 50 IS identified confirmed the proposal that there were at least two families headed by IS*1111* and IS*110*[1]. A similar comprehensive analysis of the IS*110* group is not available. IS*1111* family members recovered from available sequences could be distinguished simply from IS*110* family members via the presence of sub-terminal inverted repeats (here designated sTIR), usually 11–13 bp in length with a perfect or near-perfect match. The terminal extensions that are part of the IS but found beyond the inverted repeats (IR) were asymmetric, and later work has shown that their correct lengths are 7 bp on the left and 3 bp on the right[2,19] or occasionally 6 bp and 3 bp. In addition, the transposases aligned well with those of IS*110* and its relatives in the N-terminal catalytic (DEDD,

[1]School of Life and Environmental Sciences, The University of Sydney, University of Sydney NSW 2006, Australia. ✉e-mail: ruth.hall@sydney.edu.au; sandro.ataide@sydney.edu.au

RuvC) domain but less well at the C-terminus[1]. The presence of a non-coding region (NCR) of significant length downstream of the transposase gene (*tnp*) in all cases was noted[1], and later, it was proposed that an RNA produced from this NCR was involved in the selection of the appropriate target site[2]. However, although this feature is not found in any of the IS families encoding DDE transposases, the NCR is generally not mentioned in later descriptions of these IS[4,14,18]. Over time, both families have grown, but despite the clear differences, they were combined at some point and are currently grouped together under "IS*110* family" in the ISFinder database[20] (www-IS.biotoul.fr) and discussed together in reviews of IS sequences (e.g[18]). Indeed, as the features of and relationships within the IS*110* type have not been examined in detail, there may be further families embedded in this group.

Experimental characterisation of IS in these two families is limited. However, members of both the IS*1111* and IS*110* families have been shown to form a circular intermediate in which the IS ends are directly abutted[1,2,13,21–25]. This creates a promoter that, in a few cases, has been shown to be active[22,23]. This promoter is presumed to drive transcription of the *tnp* gene and would also transcribe the NCR. In a few cases, transposition to the appropriate target has been demonstrated for both IS*1111* family IS[1,22,23] and IS*110* family IS[13,21,26]. Though a duplication of bases in the target sequence has been claimed, in other cases no duplication was found. In fact, a duplication of target sequences does not appear to be characteristic of IS in either family, and this discrepancy likely results from incorrect identification of the ends of the IS, which should include one copy of the duplication. Further information on what has previously been reported about these families can be found in the online resource TnPedia in TnCentral[14].

Here, we have first confirmed the separation of the IS*1111* and IS*110* families, showing that the longer NCR is upstream of the *tnp* gene in IS*110* family members and examining the phylogeny of a curated set of predicted transposase sequences. We then examined the role of the NCR in IS movement and in selecting the correct target. We used several IS*1111* family members, ISEc11[22], which moves to a short linear target, ISKpn4 and ISPst6 which both target one end of certain *attC* sites of integron-associated gene cassettes[2,23], and ISPa11 which targets the REP sequences in *Pseudomonas aeruginosa* chromosomes[1]. We also examined one IS*110* relative, ISEc21, for which the target sequence was known, but its location was not, and this was determined here. We show that a seeker RNA (seekRNA) determined by the NCR found downstream of the *tnp* transposase gene in the IS*1111* ISs and upstream of *tnp* in the IS*110* IS, is essential for movement and co-purifies with the transposase. The seekRNAs were 70-100 nt and were more abundant than a longer RNA, a potential precursor, that completely includes the seekRNA. Both the seekRNA and the long form include a short region complementary to each strand of the target. In the seekRNAs of IS*1111* family IS, the top strand complement precedes the bottom strand match, but for the IS*110* family IS, the order was reversed. Hence, the mechanics of target selection may be different in the two families. Instances of natural reprogramming were identified, and we reprogrammed the system for a representative of both IS*1111* and IS*110* families to insert a mini IS carrying a cargo gene into chosen sites.

## Results
### IS*1111* and IS*110* family features
Here, we noticed that the longer NCR is upstream of the *tnp* gene in IS*110* and some of its relatives rather than downstream as in the IS*1111* type (Fig. 1a). Hence, over 350 IS listed in ISFinder (www-IS.biotoul.fr), currently under the designation IS*110* but grouped as IS*110*, IS*1111* or unknown type, were examined to locate the longer NCR. 186 had the NCR downstream of *tnp*, 123 upstream, 26 had two longer NCRs, and 17 had none. Almost all correctly classified IS*1111* family IS, i.e. those with

sTIR of sufficient length and a transposase related across the full length to that of IS*1111*, had the longer NCR downstream of the *tnp* gene. However, a few had an upstream NCR or a large NCR both up and downstream. In the remaining IS, the NCR was generally upstream, and sTIR was not present or was short and imperfect. However, some of these had a 3′-NCR. Hence, the NCR location appears to be a characteristic of the two families, albeit with some exceptions. This finding supports the conclusion that the IS*1111* family is indeed distinct from the IS*110* family and that the mechanisms used by these two families may differ significantly, particularly with regard to end recognition.

Previous studies had found that the N-terminal catalytic domain of the transposase in both IS*110* and IS*1111* relatives is related, but the C-terminal domain aligns poorly[1,13,27], and our alignments (Supplementary Fig. 1) confirmed this. While in the logo for the IS*1111* family transposases, a clear pattern of consensus residues is apparent in both the N- and C-terminal regions, only an SG (part of a G--P----SG motif) is highly conserved in the C-terminus for the IS*110* family transposases. For the IS*1111* transposases, this location is represented by SG or TG (or occasionally TA[1]) in a GL-P----S(t)G motif. Conservation of the SG residues in a G--P----S(t)G motif was noted previously[13,27], and these two aa were shown to be required for inversion of the invertible DNA segment of *Moraxella* catalysed by Piv[27], which is a relative of the IS*1111* and IS*110* family transposases[28].

A phylogeny of the transposases constructed from the set of IS listed in ISFinder, curated to include only a single representative for each group of transposases with >70% pairwise aa identity (Supplementary Fig. 2), revealed that the transposases of IS*1111* family members, classified based on the presence of sTIR of appropriate length, are clearly separated from the remaining transposases from IS*110* members. Two exceptions (marked in blue) are found in a deep branching group of transposases from IS classified as IS*110* family. A simplified phylogenetic tree (Fig. 1b) highlights some further exceptions. In the tightly clustered group of IS*1111* relatives, all but one had a downstream NCR. In ISMtsp17, the NCR was upstream, in contrast to its closest relative ISPye21, which had a standard downstream long NCR. The more diverged IS, ISHvo10 and ISAcp5, have long NCRs both up and downstream. For the IS in the IS*110* family, the NCR was found mainly upstream of *tnp*, but it is downstream in some, e.g. the group highlighted in green in Fig. 1b. Transposases represented in the dark blue group in Fig. 1b that includes IS*492*[21] and IS*621*[13] are most closely related to the Piv inversion proteins[12,28]. The phylogeny (Supplementary Fig. 2) also indicates that there are clear subfamilies within each family, and there may be further families, particularly in the current IS*110* group, granted the large range of lengths (315–450 aa) found in transposases encoded by members of this family compared to the more compact spread of lengths in the IS*1111* family transposases (330–350 aa). For example, transposases in the branch of the IS*110* family that includes the IS*492*, IS*621* and the Piv invertase (Supplementary Fig. 2) are generally much shorter at 315-330 aa than the transposases of members of the subgroup that includes IS*110*, which range from 395–455 aa. The transposases of IS in the cluster that includes ISEc21 used in this study are of intermediate size (345–385 aa).

However, there are clear similarities in the folded structures of the transposase proteins. Using AlphaFold2[29], the structures of several transposases derived from each family were modelled, and transposase TnpEc11 from ISEc11 (339 aa), as a representative for the IS*1111* family, and TnpEc21 from ISEc21 (358 aa), as a representative of the IS*110* family, are shown in Fig. 1c and Fig. 1d, respectively. The transposases all consist of an N-terminal DEDD (RuvC fold) catalytic domain[28] and a C-terminal domain with no well-characterised homologues (includes Pfam PF02371; InterPro IPR00346) separated by a coiled-coil with a variable region between the α-helices. More detailed representations of these transposases showing the configuration of the DEDD residues and the location of the SG motif at the tip of the finger in the C-terminal fold are in Supplementary Fig. 3.

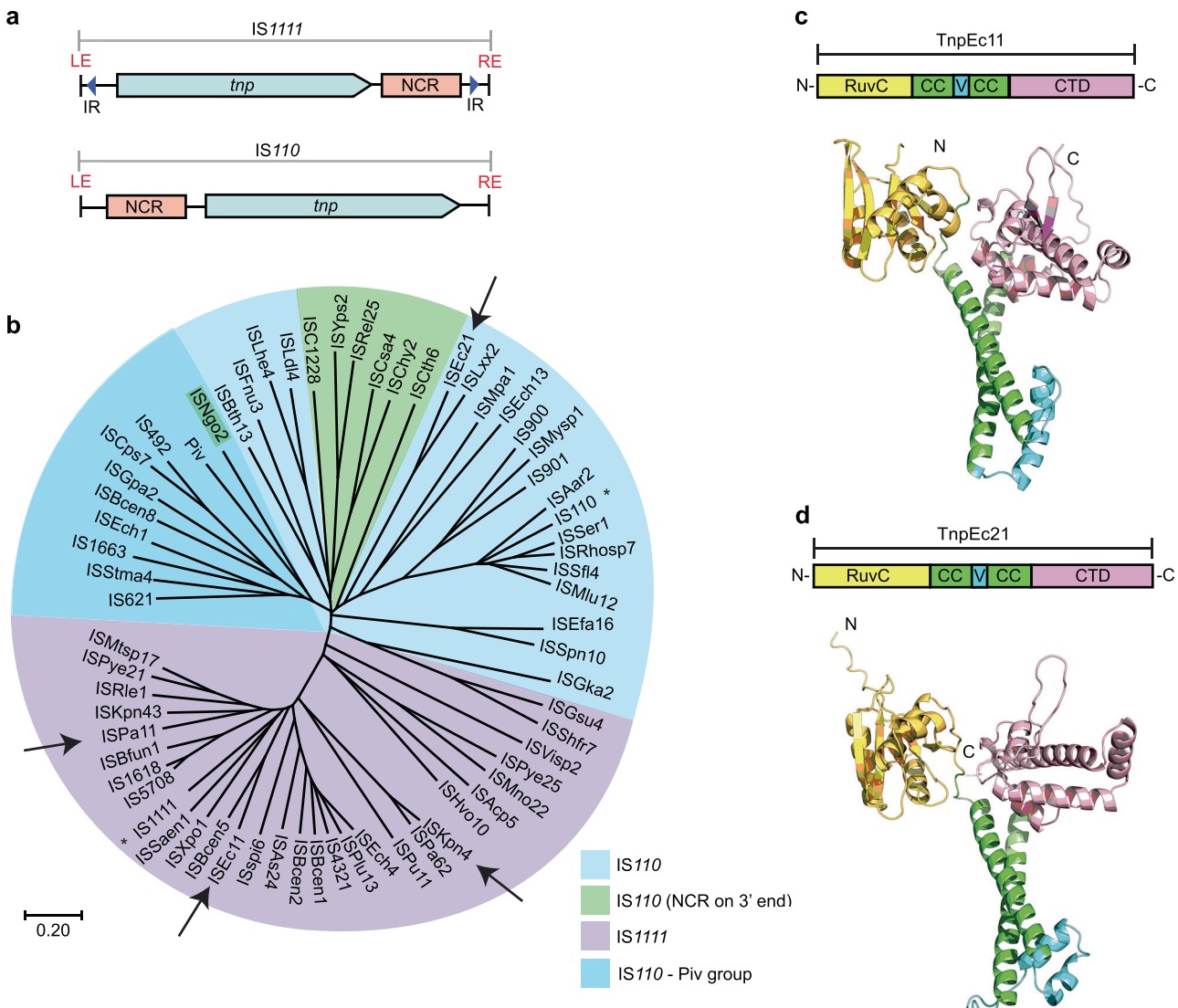

**Fig. 1 | IS1111 and IS110 family features. a** Organisation of IS1111 and IS110 family members showing locations of the longer non-coding region (NCR, pink box) relative to the transposase gene, *tnp* (blue arrow). Subterminal inverted repeats (sTIR) in IS1111 ISs are indicated by arrows labelled IR. **b** Simplified phylogenetic tree of a few transposase sequences from IS1111 and IS110 family members and Piv, rooted by the midpoint. The scale bar represents a 20% divergence in protein sequence. Founding family members IS110 and IS1111 are indicated by asterisks. Arrows point to ISs studied in this work. The colour code key is indicated. AlphaFold 2 model prediction of domain structures of **c** TnpEc11 and **d** TnpEc21 as examples of IS1111 and IS110 family transposases, respectively. The domains are RuvC (yellow), coiled-coil domain (CC; green), variable region (V; cyan) and an uncharacterised C-terminal domain (CTD; pink) (pfam: PF02371). N- and C-termini are labelled N and C, respectively. Source data are provided as a Source Data file.

The boundaries of the various domains are marked on the conservation analysis for TnpEc11 and TnpEc21 derived using the ConSurf server (consurf.tau.ac.il) (Supplementary Fig. 4). The variable region, which is composed of small α-helices and loops, is longer in the ISEc21 transposase than in that of ISEc11. AlphaFold2 also predicts the formation of a dimer via interactions between the coiled-coils placing the variable region of one monomer next to the RuvC domain of the second monomer (Supplementary Fig. 3). The AlphaFold2 modelling also predicts a tetramer as shown only for TnpEc11 in Supplementary Fig.3.

Further transposases derived from the other IS110 subfamilies were modelled, and the variation in transposase lengths seen in the subgroups of the IS110 family transposases was largely accounted for by variation in the length of the variable region. This region consists of a simple loop in Tnp492 and Tnp621 from the Piv group, and in Tnp110, there is a larger region of helices and loops than in TnpEc21. However, the precise role of the variable region is not currently known.

## The downstream NCR in ISEc11 is essential

ISEc11 is an IS1111 family member that was reported to recognise a short target (GTGAAAATACTG) and has been shown to form a circular intermediate in which an active promoter is generated at the junction and to transpose to its preferred target[22]. Here, ISEc11 (cloned in pUC19 together with approximately 100 bp of flanking sequence on each side) produced a circular intermediate that was readily detected using PCR (Fig. 2a). However, the uninterrupted target sequence that would be regenerated if the flanking sequences were always re-joined in the course of the excision process (as in a site-specific tyrosine or serine recombinase reaction) was not detected under the same conditions, indicating that the reaction is not equivalent to site-specific recombination. However, when the 3′-NCR was deleted (bp 1170–1374 of 1443 bp removed), leaving the transposase gene and right end intact, the circular form was not produced (Fig. 2a), consistent with an essential role for this region. Moreover, when the 270 bp fragment that includes the NCR (bp 1127–1397 of ISEc11 preceded by a T7 promoter

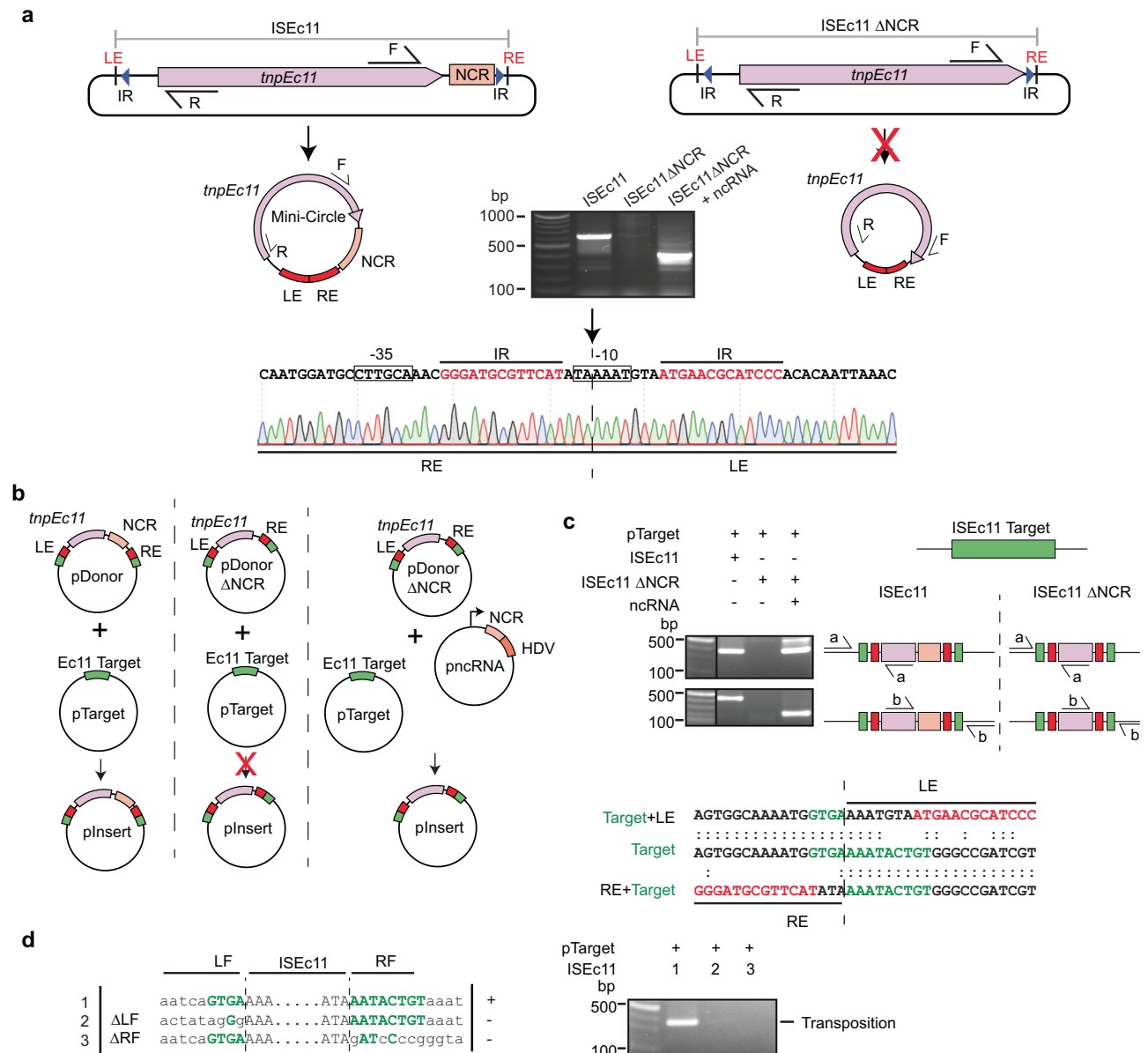

**Fig. 2 | Features of the IS*1111* family member ISEc11 required for transposition.** **a** Circular intermediate formation. The plasmids containing ISEc11 or ISEc11ΔNCR lacking the non-coding region (bases 1170–1374 of 1433), flanked by the ISEc11 target, are shown above predicted circular intermediates. The red cross on the arrow indicates the reaction did not occur. Primers facing outward from the *tnp* gene (pink arrow) used to detect circular intermediates are indicated by arrows labelled R and F above. In the centre, a gel shows the PCR products for ISEc11 and for ISEc11ΔNCR complemented by pncRNA (structure shown in Fig. 2b). Sanger sequencing of the R-F PCR product formed for intact ISEc11 is shown below with the −35 and −10 motifs of the promoter formed by joining the left and right ends (LE and RE) boxed and the sTIR in red type. The junction is indicated by a vertical arrow and dashed line (*n* = 2 transformations). **b** Transposition in vivo. Plasmids present are shown as schematics above the vertical arrow with the predicted product, labelled pInsert, below. The ISEc11 target is green, the right and left ISEc11 ends are red *tnpEc11* is a large pink box, and the NCR is a smaller pink box. In pncRNA, the

NCR is preceded by a T7 promoter (bent arrow) and followed by an HDV ribozyme (orange box). The red cross indicates the reaction was not detected. **c** PCR detection of in vivo transposition. On the left, a gel showing PCR amplicons was detected. Primer pairs (**a** and **b**) used to detect transposition are indicated on a schematic of potential products. Colours as in (**b**) and the thin black line indicates the vector-specific sequence in pTarget. Sequences of the PCR products for ISEc11 are shown below compared to the target sequence with specific target sequences in green and IR bases in red. The junction between the ends and the target is indicated by the vertical dashed line (*n* = 2 transformations). **d** Transposition with altered target sequences flanking ISEc11 and in pTarget. Sequences tested are on the left with consensus target bases green and the boundaries between IS and target indicated by dashed lines. PCR products formed using the primer set in Fig. 2c are shown in the gel to the right (*n* = 2 transformations). Source data are provided as a Source Data file.

and followed by an HDV ribozyme; pncRNA in Fig. 2b) was supplied in a separate plasmid, the circular form was again produced (Fig. 2a). Likewise, transposition of ISEc11 to its preferred target in a separate plasmid required the presence of the NCR in cis or in trans (Fig. 2b). In the product, the IS was inserted into the target at the appropriate

position and in the correct orientation, and as expected for IS*1111* family members no additional bases were generated (Fig. 2c). Note that, previously, a 4 bp target site duplication was claimed for this IS[22] but here one copy of the 4 bp was assigned to within the IS at the left end to form the required 7 bp extension at that end. Replacement of

target bases flanking the donor on either side prevented both circle formation and transposition (Fig. 2d), demonstrating that intact target-derived sequence on both sides of the donor is needed for the formation of the circular intermediate.

## IS*1111* family transposases co-purify with NCR-derived RNA

The ISEc11 transposase, TnpEc11, was expressed (using a HisMBP and Strep TagII fusion), either with or without the full-length ISEc11 present in the same cells to supply the NCR, and then purified. The yield of

TnpEc11 was poor in the absence of the NCR region but improved when the complete ISEc11 (preceded by a T7 promoter) was present to supply the NCR (Fig. 3a). The TnpEc11 expressed in the presence of the NCR purified with a nucleic acid and two distinct bands were clearly present. Their sizes are estimated to be approximately 80 and 150 nt. The nucleic acid was digested by RNase but not by DNase (Fig. 3b). The RNA extracted from the affinity-purified RNP complex (Fig. 3c) was sequenced, and the reads were aligned with the ISEc11 sequence where they mapped to the 3′-NCR (Fig. 3d). The sizes of the bands seen in the

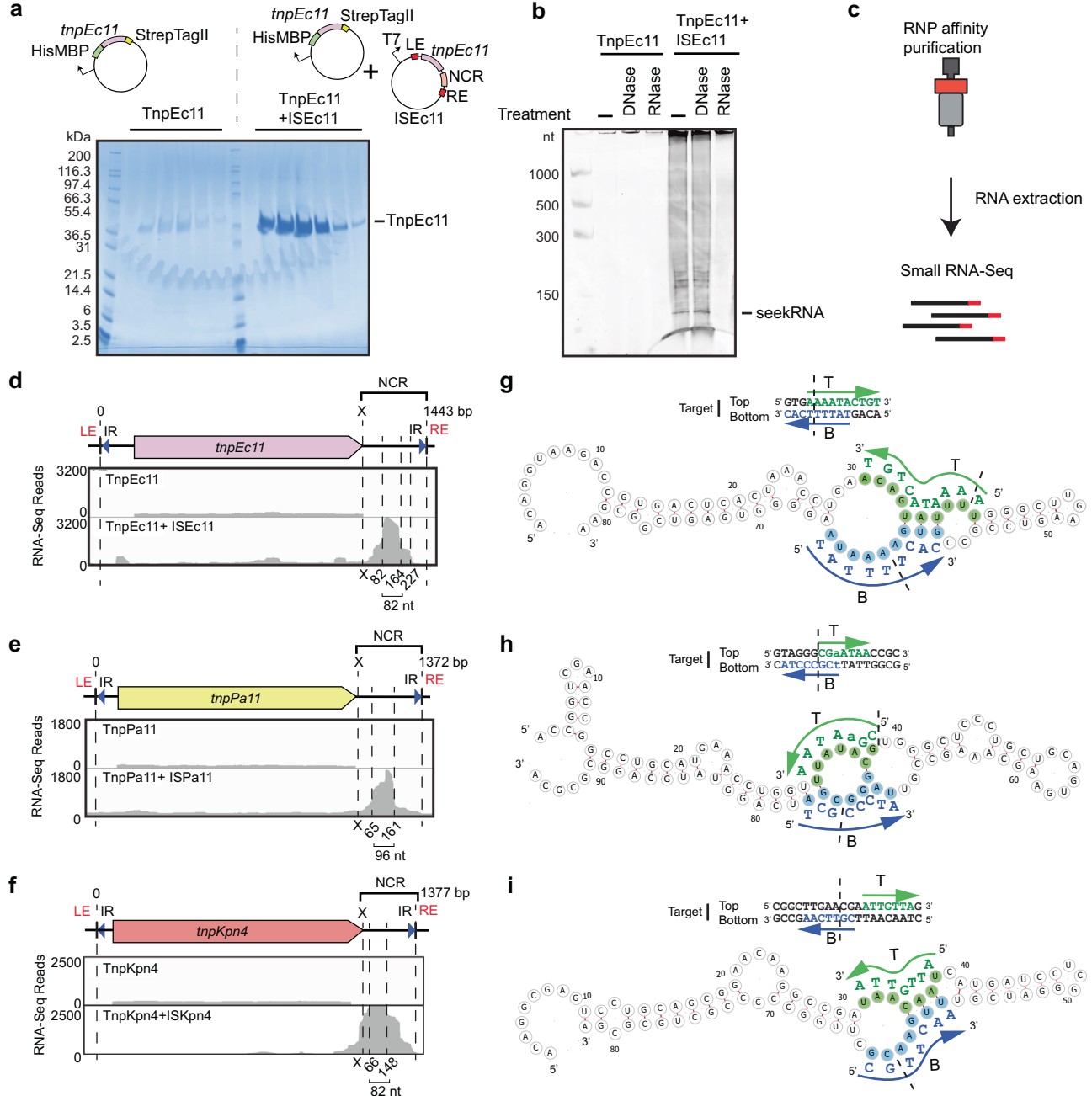

**Fig. 3 | Characterisation of seeker RNA bound to IS*1111* family members. a** SDS-polyacrylamide gel electrophoresis (PAGE) of TnpEc11 purified alone or in the presence of ISEc11, showing TnpEc11 yields were greatly enhanced by the presence of the NCR (*n* = 2 purifications). **b** SYBR GOLD-stained denaturing PAGE of TnpEc11 purified alone or in the presence of ISEc11 after digestion with DNase or RNase (*n* = 2 purifications). **c** Schematic showing the RNP and RNA purification and RNA sequencing steps. **d** Small RNA-seq reads mapped to ISEc11 with the sizes of the long seekRNA and the peak seekRNA below, together with their location relative to

the *tnp* termination codon (x). **e**, **f** As in **d** for ISPa11 (**e**) and ISKpn4 (**f**). **g** Folded structure prediction of the ISEc11 peak seekRNA with the target logo above. Sequences that base pair with the top (T) strand of the target are highlighted in green, and the bottom (B) strand of the DNA target is blue. Arrows indicate the 5′–3′ of the target DNA strand, and dashed lines indicate the insertion point of the target site. **h**, **i** As in **g** applied to ISPa11 (**h**) and ISKpn4 (**i**). Source data are provided as a Source Data file.

denaturing gel correspond to the sizes of the peak RNA of 82 nt (Fig. 3d) and the longer RNA (154 nt; Supplementary Fig. 6a). Hence, specific RNAs transcribed from the essential NCR were associated with the transposase. As inspection of the IS sequence did not reveal any promoter other than the one formed by bringing the ends together in the circular intermediate, the predominant shorter RNA would be derived from the longer form.

To confirm that this property is found more widely in IS*1111* family members, two further IS that target different sites were examined. ISKpn4 and ISPa11 have previously been shown to form a circular intermediate and are found at a specific location in a potentially folded structure, an *attC* site of integron-associated gene cassettes for ISKpn4[2] and a *P. aeruginosas* REP for ISPa11[1]. As for ISEc11, TnpKpn4 and TnpPa11 were both purified with associated RNA (Supplementary Fig. 5) and the long and short RNAs observed both mapped to the region downstream of the *tnp* gene (Fig. 3e, f).

## A seekRNA finds the target

To locate the target-determining region or regions, we searched in the sequence of the predominant (or peak) 82 nt RNA associated with TnpEc11 for short sequences that could base pair with the forward and/ or the reverse strand of the target sequence. Two short segments, one complementary to the forward or top strand (T in Fig. 3g) and one complementary to the reverse or bottom strand (B in Fig. 3g), were found, and their location in the folded structure of the RNA, predicted using MXfold2[30], is shown in Fig. 3g. These matches overlap in the target (Fig. 3g). As these matches explain the ability to select a specific target, the shorter NCR-derived RNA was named a seeker RNA or seekRNA. The folded structure for the longer RNA (long seekRNA) is in Supplementary Fig. 6a. The sequences of the seekRNA corresponding to the predominant peak for ISPa11 (96 nt) and ISKpn4 (82 nt) were also examined, and matches located in the predicted folded structure (Fig. 3h, i). Again, two short segments complementary to the forward and the reverse strand of the target sequence were found in the sequence of the predominant shorter band and were in the same order as in ISEc11 seekRNA, namely top strand match located 5′- to the bottom strand match. Predicted folds for the corresponding long seekR-NAs are in Supplementary Figs. 6 and 7.

ISPst6 inserts into the same position in a target the same or very similar to the ISKpn4 target[23], and TnpPst6 is 86% identical (92% similar) to TnpKpn4. ISPst6 also produced a seekRNA that purified with the transposase (Supplementary Fig. 5) and the ISPst6-derived 86 nt seekRNA contained the same matches as the ISKpn4 seekRNA (Supplementary Fig. 7). We also examined the NCR of ISPa25 that also targets this site but encodes a significantly diverged transposase (TnpKpn4 and TnpPa25 are 46% identical; 60% similar)[2]. The region in the DNA sequences where they most closely match includes the region corresponding to the seekRNA of ISKpn4 (Supplementary Fig. 7). The predicted seekRNA was found to include the same stretches of sequence matching the target and assume a very similar fold (Supplementary Fig. 7). Hence, the seekRNA does not appear to be confined rigorously to interaction only with a specific group of very closely related transposases (>70% identity).

## The seekRNA is programmable

ISEc11 and ISXne4 represent an example of the natural re-direction of an IS via replacement of the target detection region of the seekRNA. When the predicted seekRNA of ISXne4, a relatively close relative of ISEc11 (TnpEc11 and TnpXne4 are 72% identical; 88% similar), as compared to the seekRNA of ISEc11 (Supplementary Fig. 8), a high level of identity was observed in the stem region but the remainder diverged, and the target-matching bases in the ISEc11 seekRNA were not found (Fig. 4a). Examination of the location of the few known copies of ISXne4 revealed that it was surrounded by a different sequence (only 3 of 13 bp are the same) and that this target matched the differing

regions (Fig. 4a). This confirms that the two target-matches have been correctly identified and demonstrates re-direction.

To re-program the IS to move to a different target, it was necessary to change both the sequences corresponding to the target that flank the donor IS and the target matching sequences in the seekRNA. For ISEc11, two new targets were tested using the movement of a mini IS containing the mCherry gene bounded by the IS ends (LE 50 bp, RE 46 bp) and flanks containing the target (Fig. 4b) but without an upstream promoter. The TnpEc11 and the appropriate long (154 nt) seekRNA were supplied in the donor plasmid. Movement to targets (Fig. 4c and Supplementary Fig. 9) that were preceded by a T7 promoter enabled expression of the mCherry, measured by flow cytometry (Supplementary Fig. 10). Using the wildtype target and seekRNA, transposition occurred at a frequency of about 15% (Fig. 4d). When the portion of the target that flanks the IS on the right was altered and the corresponding changes were made in the seekRNA (see Supplementary Fig. 9 for details), transposition of the mCherry to the new M1 target occurred at about 23% frequency. In the second case, the target on both sides of the donor mCherry mini IS and corresponding positions in the seekRNA were altered, and again the transposition frequency to the new M2 target was 15%. Hence, the long seekRNA could be programmed to move the IS to a different location. Using the same assay, the short 82 nt seekRNA (the peak in Fig. 3d) was also tested, and the mCherry mini IS moved even more efficiently (42% transposition; Fig. 4d, bottom line), indicating that this length is sufficient to support IS movement.

## Moving cargo

To demonstrate that a cargo gene or genes can be moved by these IS, the *catA1* chloramphenicol resistance gene with its upstream promoter and ribosome binding site (adding 745 bp) was introduced at the start (after bp 50) and at the end (after bp 1397 of 1443) of ISEc11 (Fig. 5a). In both cases, movement was detected. The central portion of ISEc11 was also replaced by the *catA1* gene with an upstream promoter leaving the ends intact. When the transposase and NCR were supplied in trans, again movement was detected (Fig. 5b) as seen for mCherry, indicating that the system can be harnessed to mobilise different cargos.

## ISEc21 from the IS*110* family: similarities and differences

ISEc21 is an IS*110* family member with an upstream NCR (Fig. 6a). The target listed in ISFinder was confirmed computationally and here determined to be a linear target after it was traced to a conserved region within certain IS*3* family members (e.g., ISCfr6, ISEc92, ISEc93), namely the codons for and surrounding the second D of the DDE in the catalytic domain of the transposase (Supplementary Fig. 11). As for ISEc11, ISEc21 formed a circular intermediate in which the ends were abutted and a promoter generated (Fig. 6a). When the upstream NCR was deleted (bp 20–150 removed), circle formation no longer occurred (Fig. 6a). The NCR was also needed in cis or in trans (bp 8–179 in ISEc21 preceded by a T7 promoter and followed by an HDV ribozyme) for transposition (Fig. 6b, c) and movement into the target supplied at the correct position and in the correct orientation without generating a duplication of bases at the target site was detected (Fig. 6d). When the number of target bases matching the consensus was reduced to only 3 on the right or to 5/6 on the left and 5 on the right of the donor IS, both minicircle formation and transposition still occurred (Fig. 6e). However, removal of further bases on each side abolished movement, again indicating the importance of the presence of the two parts of the target sequence surrounding the donor IS for recognition of the IS ends and to bring them together in the minicircle.

The TnpEc21 transposase was purified with RNA (Supplementary Fig.5) that was recovered and sequenced. The sequences mapped to the upstream NCR (Fig. 7a, Supplementary Fig. 12). Again, a shorter seekRNA was more abundant than the long seekRNA (see Supplementary Fig. 12 for the fold of the long seekRNA). Bases matching both

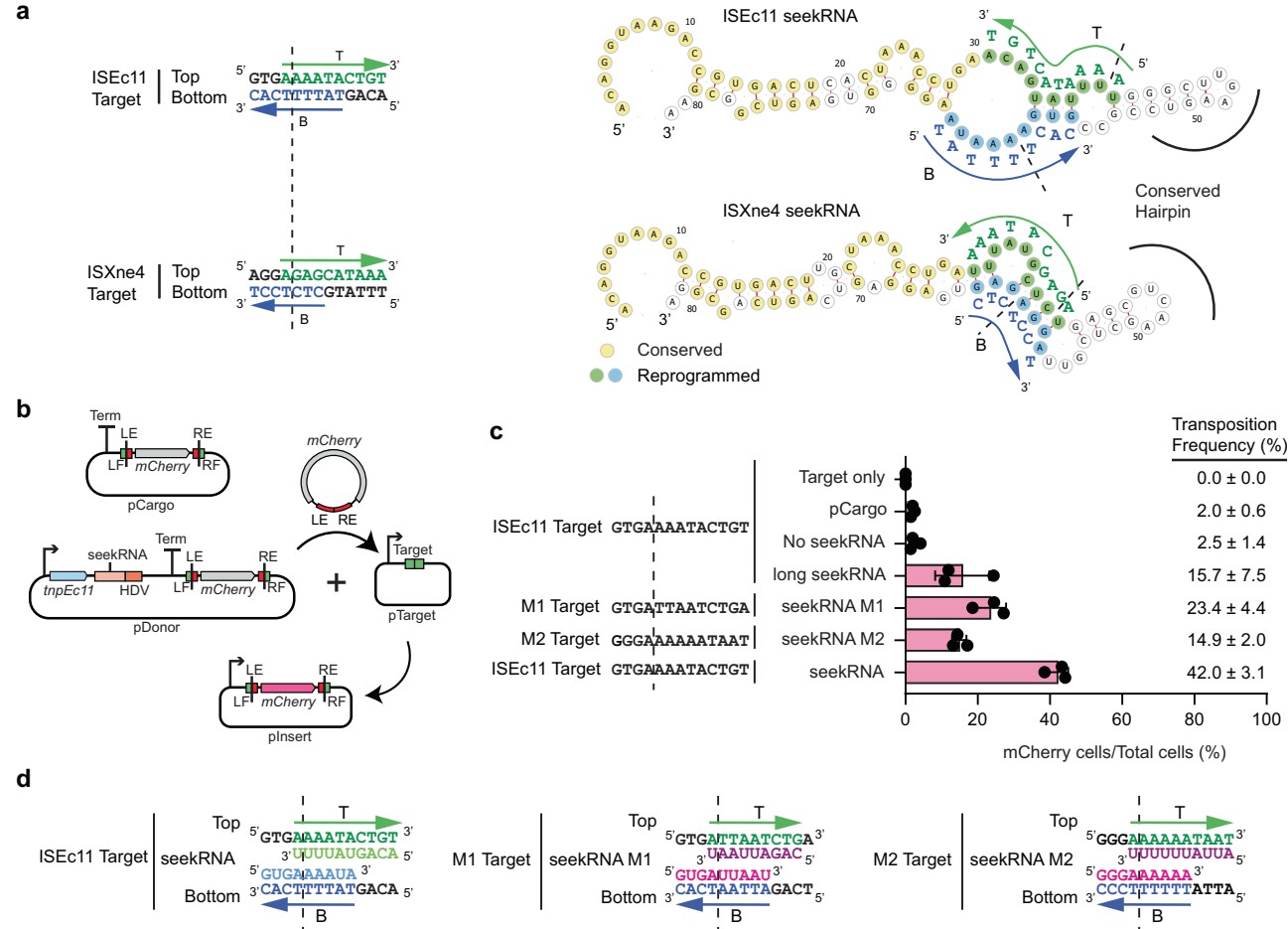

**Fig. 4 | Natural and experimental reprogramming to recognise a new target site. a** Features of ISEc11 and ISXne4 indicating natural reprogramming. The target sites for ISEc11 and ISXne4 are shown on the left; a dashed vertical line indicates the IS insertion point. Bases complementary to ones in the corresponding seekRNA are green in the top strand and blue in the bottom strand. On the right, folded structure predictions for ISEc11 and ISXne4 seekRNAs with nucleotides conserved in both yellow, nucleotides complementary to their target site in green or blue and matched target bases in adjacent capitals. The corresponding bases in the target are shown in green and blue. Arrows indicate the 5′–3′ direction. **b** Schematic of the mCherry reporter assay. The pDonor plasmid contains a T7 promoter (bent arrow) preceding *tnpEc11* (blue box and the natural or re-programmed seekRNA (pink box) followed by an HDV ribozyme (orange box) with a strong T7 terminator at the 3′ end. The mini IS contains mCherry without a promoter surrounded by the left IS end (LE) plus left flanking (LF) sequence and the right IS end (RE) plus right flanking sequence (RF); the target is green, and the ends

are red. pTarget includes a target (LF and RF abutted) preceded by a T7 promoter (bent arrow). **c** In each assay, the LF and RF in the donor and target plasmids are identical and appropriate to the matches in the seekRNA. Transposition efficiency of the mCherry mini IS by TnpEc11 with the natural (ISEc11) or reprogrammed long (M1 and M2) or the ISEc11 seekRNA. The target sequences used are shown with a vertical dashed line marking the insertion point. Black dots indicate values of *n* = 3 independent biological replicates, and the pink bar corresponds to the mean value with error = SD. Transposition frequencies are on the right of the displayed dots. **d** Sequences of the ISEc11 target and modified targets (M1 and M2). The base-pairing sequences present in the seekRNA are shown below the top strand target sequence (green) and above the bottom strand target sequence (blue). Arrows point to the 3′ of the DNA target site in both strands T (top) and B (bottom). The dashed line indicates the insertion point of the target site. Source data are provided as a Source Data file.

strands of the target were found in the 74 nt seekRNA (Fig. 7a). How-ever, in contrast to the situation in the seekRNAs from all of the IS*1111* family members tested here, the top strand match (T in Fig. 7a) is located 3′- to the bottom strand match (B in Fig. 7a).

A case of natural redirection was also found in an ISEc21 relative. The transposase of ISMch6 is 72% identical (85% similar) to TnpEc21, but the IS is found in a modified target. Comparison of the DNAs of the NCR revealed three altered bases in the target-defining region for the bottom strand (Supplementary Fig. 12) and the similarities between the predicted folded short seekRNAs for ISEc21 and ISMch6 are shown in Supplementary Fig. 12. Although only a single copy of ISMch6 was found in sequences available in GenBank, the differences found in the sequence surrounding ISMch6.

Using the assay shown in Fig. 7b, a 58 nt RNA that is 16 nt shorter than the 74 bp ISEc21 seekRNA was found to be sufficient to support

transposition. Movement of a promoterless mCherry in a mini IS, surrounded by the IS ends (LE 29 bp, RE 16 bp) with 15 bp flanks car-rying the ISEc21 target site, moved to its usual target site (Fig. 7c). The system was also re-programmed. One of the target-matched regions in the shortened 58 nt ISEc21 seekRNA 2 form was changed to detect a different sequence, and mCherry was surrounded by the outer IS ends now flanked by the new target sequence (Fig. 7 d Supplementary Fig. 13). Transposition occurred when the target site surrounding the donor was the same as the target offered and seekRNA included appropriate nucleotides to detect that target (Fig. 7d).

**Many IS, many different targets**

The most unusual feature of the IS*1111* and IS*110* families is that the target recognised by each IS or group of IS is not the same. A com-putational pipeline to identify a consensus target site by searching the

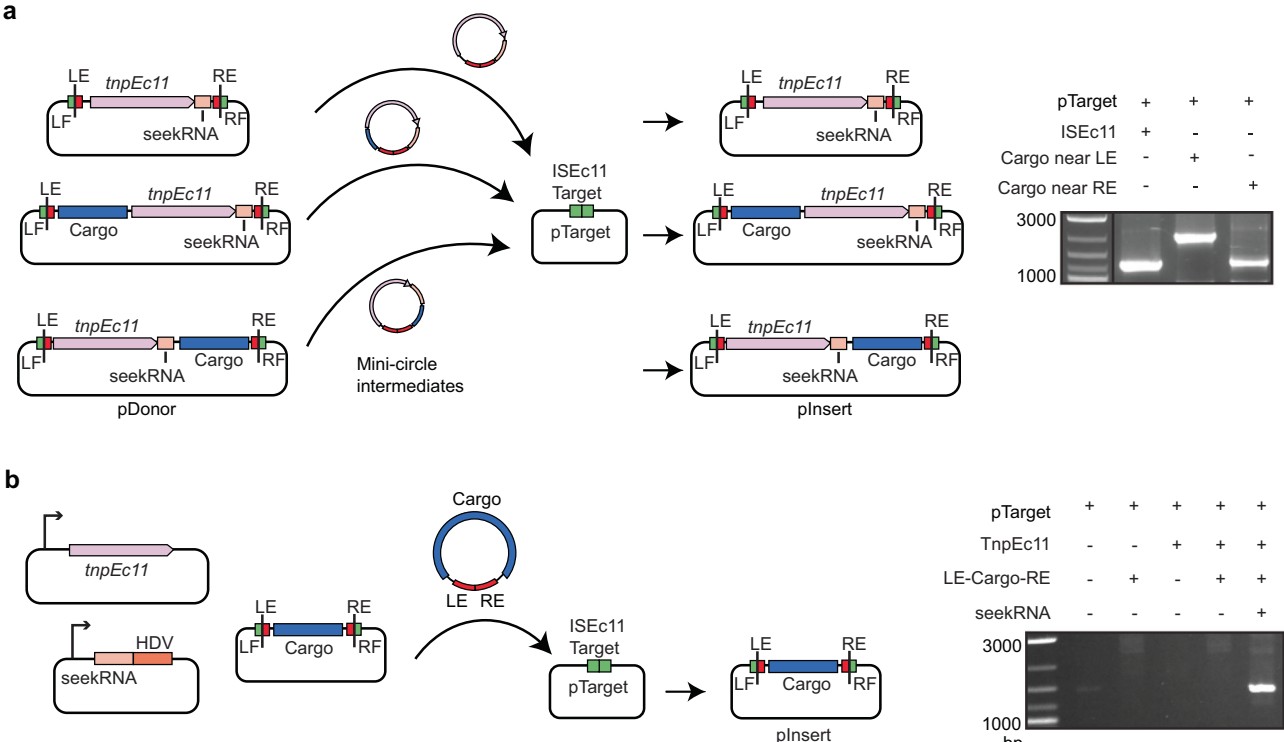

**Fig. 5 | ISEc11 can carry cargo. a** Schematic of transposition assay for ISEc11 carrying cargo. The pDonor plasmids include ISEc11 or ISEc11 with the *catA1* gene (the blue box labelled cargo) inserted upstream (50 bp from the start of the IS) or downstream (46 bp from the end of IS) of the *tnp* gene (large pink box). pTarget included the ISEc11 target (green box). The NCR (small pink box and IS ends (red boxes) and flanks including the target (green boxes) are also indicated. The minicircle intermediate and predicted transposition products are also shown. In the right panel, gel showing PCR products detected using primers in pTarget and in *tnp* (primers and in Fig. 2c) demonstrating movement. **b** Schematic of mini IS

transposition assay showing the three compatible plasmids are present. The mini IS contains the *cat* cargo flanked by 84 bp from the left IS end (LE) and 71 bp from the right end (RE) and is flanked by target sequences. The *tnpEc11* (pink box) and the long seekRNA followed by an HDV ribozyme (light and dark orange boxes), each under the control of the T7 promoter (bent arrow), are in separate plasmids. Other colours are as in a. Movement of the mini IS into a fourth plasmid, pTarget containing the ISEc11 target, was detected by PCR (right panel) using a primer in *catA1* and a primer in pTarget and PCR products were sequenced. Source data are provided as a Source Data file.

sequences available in the GenBank nucleotide database was developed and applied to over 300 IS sequences found in ISFinder under the IS*110* family (https://github.com/AtaideLab/Targets)[31]. The strategy used identified the total of all copies of each IS present in all sequences found in the GenBank non-redundant database and then joined the flanking sequences (200 bp at each end). Redundant IS copies that were in the same broader position were removed so that only unique locations were considered further (Fig. 8a). Thereafter, a consensus logo was developed for 20 bp on each side of the IS. There was enormous variation in the number of unique locations found, ranging from one unique insertion event to over 7000 unique events for IS*1663* from *Bordatella pertussis* (Fig. 8b, c). The logos obtained from the pipeline were the same as those reported previously[1] for the IS*4321* target (here with nearly 900 insertions) and ISPa11 target (here with over 1000) and for several other cases. For these, the junction between the ends and the target had been found or confirmed via comparisons with uninterrupted targets[1] and finding the same targets via the pipeline confirms the accuracy of the approach. A few logos for IS from each family are shown in Fig. 8d.

However, the accuracy of this approach and the logos produced is limited by the number of unique insertions and by the accuracy of end positions in entries in ISFinder. As only one or two locations were found for 110 IS, logos could not be generated for over 30% of the IS in ISFinder. Cases yielding no logo or a logo for only one side of the IS also appeared to be common, likely due to inaccuracy in the recorded IS ends, e.g. the target may have been included with the IS. Although in some of these cases, a consensus was not generated because there

were insufficient different locations, in others, there were several to many locations, for example, Hvo9 was found in four unique locations (Fig. 8e). Another example is IS*621* with 920 unique locations (Fig. 8b) but no clear consensus for the surrounds (Fig. 8e). In this case, it was possible that there are multiple versions of the IS (>95% identical) that target different sites. However, the same logo was found when the stringency in searches was raised to 100%. Clearly, manual curation will be needed to address this unexpected finding.

From this analysis, it appears that, in a number of cases, the ends listed in ISFinder may be incorrect and need adjustment. This likely reflects, at least in part, the difficulty of finding ends of IS that do not have TIR or sTIR. For example, some early entries record adjacent duplications, one of which should be in the IS, and this leads to a duplication in the target logo, as can be seen in the IS*492* logo. In other cases, a conserved target was only seen on one side, indicating that conserved bases on the other side may have been included within the predicted IS boundaries. A few early entries lack the terminal extensions found in IS*1111* family IS, and the extension bases appear in the predicted target for IS*1533*, which is the most abundant IS*1111* family member (Fig. 8b). Resolution of these issues will require identification of the uninterrupted target(s), which could not be simply achieved in the pipeline developed here, or a sequence of the circle junction which must be obtained experimentally.

## Discussion

Here, we have shown that, as we predicted previously, IS in the IS*1111* family that are abundant in bacteria and archaea use an RNA that we

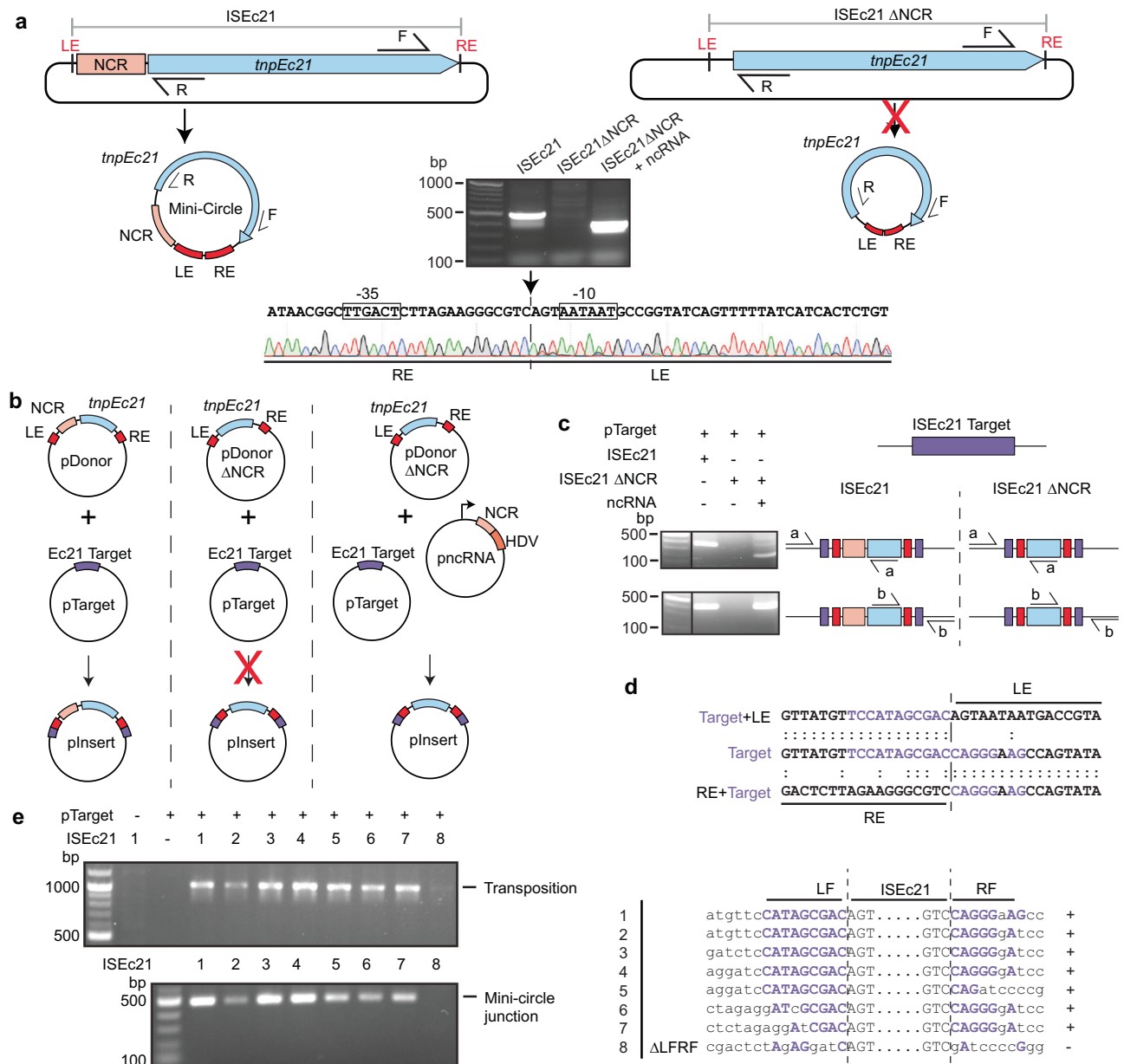

**Fig. 6 | Features of the IS110 family member ISEc21 required for transposition.** **a** Minicircle formation. The plasmids containing ISEc21 and ISEc21ΔNCR, lacking the non-coding region (bases 20–150), are flanked by the ISEc21 target. The *tnpEc21* gene is a blue arrow and the NCR a pink box. The predicted circular intermediates are below, with the ends indicated in red. A red cross on the arrow indicates when the reaction did not occur. The structure of the ncRNA plasmid used to complement ISEc21ΔNCR is shown in (**b**). Outward-facing primers R and F in the *tnp* gene used to detect circular intermediates are indicated. The gel shows the R-F PCR products. The sequence of the PCR product formed for intact ISEc11 is shown below with the −35 and −10 motifs of the promoter formed by joining the left and right ends (LE and RE) boxed and the junction indicated by a vertical arrow and dashed vertical line (*n* = 2 transformations). **b** The in vivo transposition assays. The plasmids shown in a. are combined with a compatible plasmid (pTarget) containing the ISEc21 target (purple). The ISEc21ΔNCR plasmid with pTarget and a third plasmid

carrying the NCR (light orange box) followed by the HDV ribozyme (orange box) and preceded by a T7 promoter (bent arrow) are shown. The red cross on an arrow indicates the product (pInsert) below was not formed (*n* = 3 transformations). **c** Detection of transposition. Primer sets a and b complementary to the target plasmid, and the *tnp* gene at each end of the IS is shown schematically on the right, with the PCR products shown in the gel on the left. Colours are as in (**b**) (*n* = 2 transformations). **d** Sequence of PCR products containing the LF/LE and RE/RF junctions compared to the target (purple bases are the target site). The insertion point is indicated by a vertical dashed line. **e** Alteration of flanking target sequences. The sequences in the LF and RF are shown on the right. Conserved bases in the target site are in purple capitals, and both adjacent bases and bases altered in the target are in black lowercase. Gel showing PCR products indicating transposition and minicircle formation for each sequence tested are shown on the left. Source data are provided as a Source Data file.

have named the seekRNA to detect their preferred target and take up the correct position and orientation in their specified target. We also showed that a member of the IS110 family also uses a seekRNA. In all cases, the seekRNA was essential and purified with the transposase as an RNP complex. It provides a flexible target site selection mechanism, and natural re-programming was detected among the relatives

of the IS examined here. However, the specifics of how target recognition occurs may be different in the two families as the regions that match the two strands of the target are in reverse order in the IS1111 type seekRNAs compared to the IS110 type seekRNA. Though a large 145–170 nt long seekRNA was transposase associated, the smaller, more abundant seekRNA of 70-100 nt contained the target

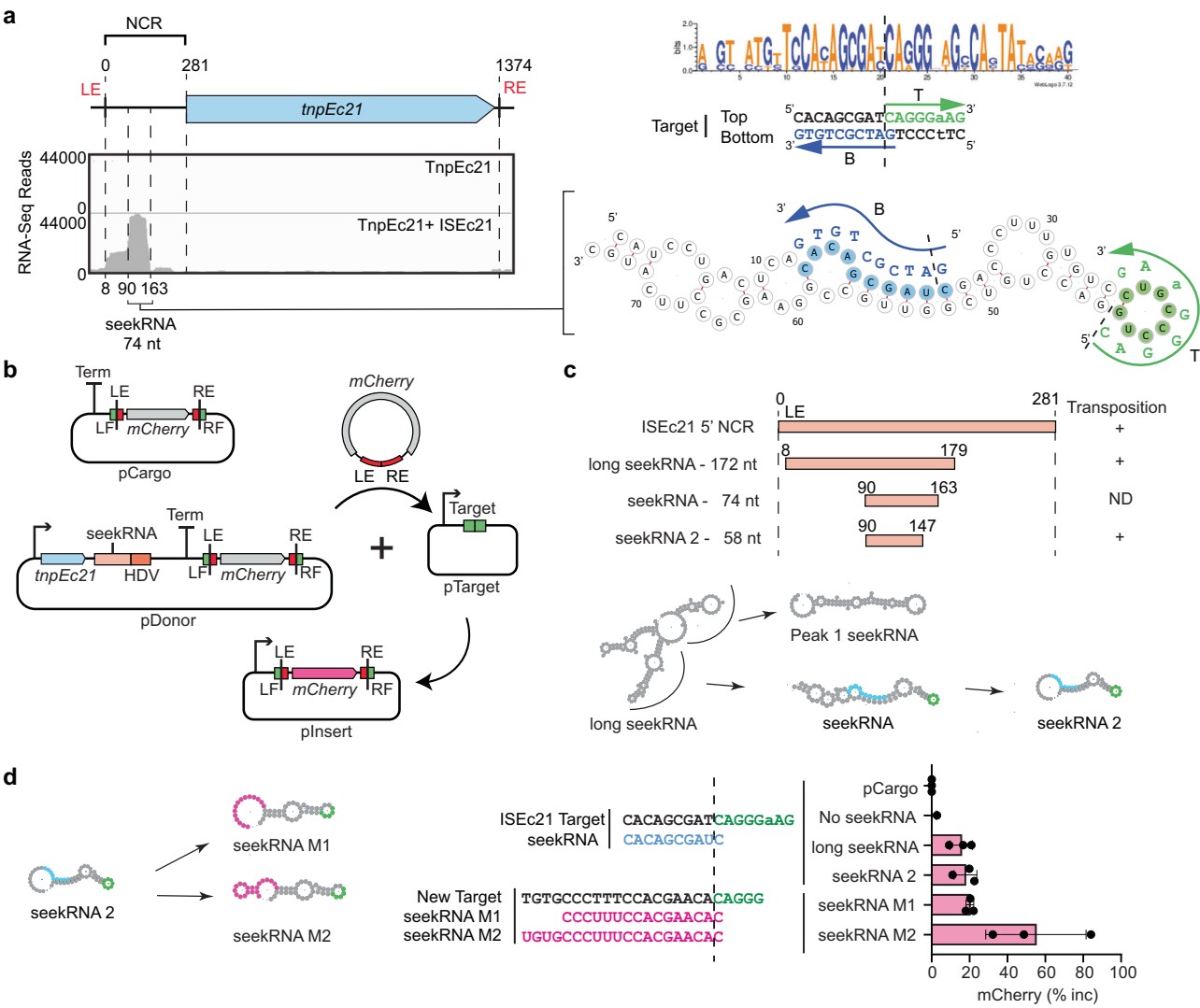

**Fig. 7 | Characterisation and experimental reprogramming of ISEc21. a** Mapping the TnpEc21 associated RNA. Small RNA-seq results from RNA purified from the TnpEc21 RNP complex aligned with the ISEc21 sequence shown schematically above. Positions of RNAs are shown together with the size of the seekRNA. On the right, predicted folds for the seekRNA with sequences complementary to the target (shown adjacent in capitals) in green and blue for top T and bottom B strands, respectively, and the arrow indicates 5′–3′. The target consensus logo is at the top, with the top and bottom strands of the target below with bases complementary to the seekRNA blue and green and arrows indicating 5′–3′. The dashed line indicates the insertion point. **b** Schematic of mCherry mini IS reporter assay. pDonor contains *tnpEc21* (blue arrow) and the ISEc21 seekRNA (light orange) followed by an HDV ribozyme (orange) expressed from a T7 promoter (bent arrow), then a terminator (Term) and a mini IS containing mCherry (grey arrow) surrounded by ISEc21 ends (LE and RE; red boxes) and a target (LF and RF; green boxes). The

mCherry gene lacks an upstream promoter. pTarget contains a target preceded by a T7 promoter. The predicted minicircle and transposition product are also shown as well as the pCargo control containing only the mini IS. **c** NCR-derived RNAs. Lengths and positions of NCR-derived RNAs tested with the outcome for transposition (+ or ND), detected by PCR and confirmed by sequencing, on the right. Predicted folds for the seekRNA forms are shown below. **d** Reprogramming. New sequences are in the target-complementary regions of the shortened seekRNA, in the target and in the LF and RF of the mini IS. Transposition was detected by PCR and sequencing of the pInsert boundaries and detected using mCherry fluorescence. Data are presented as dots of *n* = 3 independent biological replicates with the mean corresponding to the pink bar and error = SD, and calculated as % increase of mCherry fluorescence per cell as described in the "Methods". The Source data are provided as a Source Data file.

recognition regions and was shown to be sufficient for the transposition of ISEc11. An even shorter region was found to be sufficient for the ISEc21 movement. Although the precise role of the long seekRNA remains to be established, the transposase mRNA and long RNA would be co-transcribed from the promoter generated when the IS ends are brought together, and further processing, likely by the transposase, would be needed to generate the final seekRNA form. In addition, for movement to occur, it was necessary for the donor IS to be surrounded by the correct target, indicating that the seekRNA is involved in the excision of the IS to form the circular intermediate as well as in insertion into the target. Granted the difference observed between the IS*1111* and IS*110* seekRNAs and the known difference

relating to the presence of sTIR in the IS*1111* family IS, we conclude that each family was so distinct that it cannot be assumed that the specifics of the movement mechanism are the same in both families.

The IS in the IS*1111* and IS*110* families encode an unusual DEDD transposase type, and structure predictions using AlphaFold2 revealed the domain structure. Though similar overall, there were significant differences in the variable region of IS*1111* and IS*110* ISs, and this region appears to interact with the catalytic (DEDD/RuvC) domain in the dimer. How the seekRNA interacts with this protein and which domain binds to the IS ends are not known, and further in vitro and structural studies will be needed to resolve these

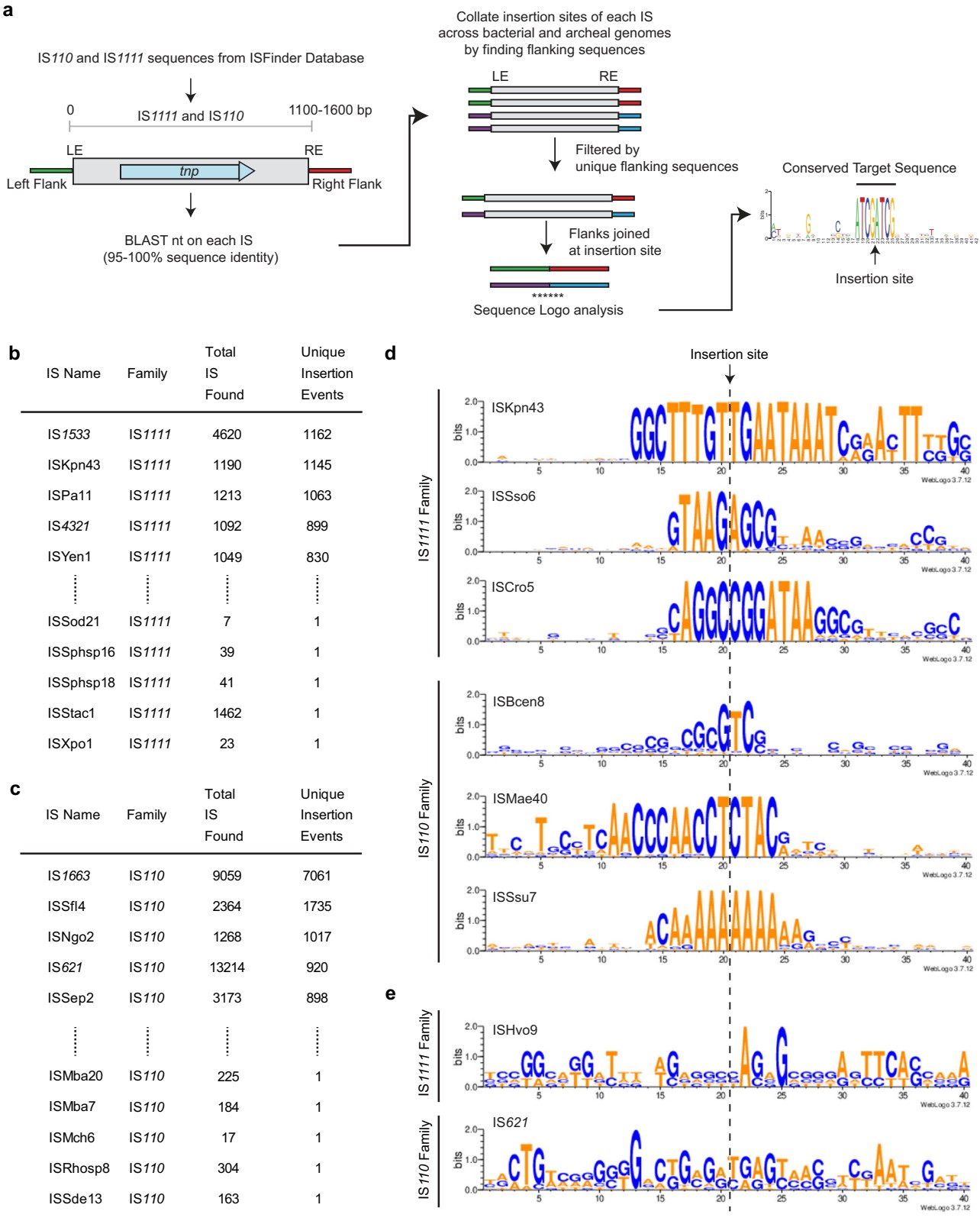

**Fig. 8 | Identification of the target site of individual IS*110* and IS*1111* family members. a** Schematic of the pipeline and scripts used to determine the target sites of the IS*110* and IS*1111* family members as listed in ISFinder. The script and documentation, along with a demo sample, are available at https://github.com/AtaideLab/Targets/ [31]. Prevalence of top and bottom 5 ISs from **b** IS*1111* family and **c** IS*110* family ordered by number of unique insertion events and showing number of copies and unique insertion events. **d** WebLogos representing the conservation in the target sites of members of IS*110* and IS*1111* families showing varied length and sequence. **e** WebLogos of examples of poor target site conservation. Logos for one IS*1111* and one IS*110* family. Are shown Source data are provided as a Source Data file.

questions. Likewise, the processes involved in generating the circular IS form without generating an equivalent re-joined target remain to be established.

The importance of systems to reliably introduce large segments of DNA to a specified location in a directional manner has recently been highlighted[32–34]. The potential for the systems studied here to be exploited for genetic engineering and biotechnology applications is supported by our finding that a cargo flanked by IS ends can be moved and that the target site can be changed by altering both the target-matching stretch in the seekRNA and the sequence flanking the donor. However, the orientation specificity of IS movement that enables orientation-specific insertion will need to be accounted for when selecting a new target. Although future work will be needed to determine the longest and shortest lengths of cargo insert that can be efficiently moved, a clear advantage of the systems reported here over others studied to date[35–38] is the fact that only a single protein of modest size (Tnp are about 340 aa for IS1111s and 315–450 for IS110s) plus a short seekRNA is required.

## Methods

### Phylogenetic analysis and modelling of IS110 and IS1111 family members

The protein sequences of the transposases encoded by annotated members of the IS110 and IS1111 families (349 sequences) were downloaded from ISfinder database[20] and curated as follows. Full-length protein sequences that contained the DEDD motif and correct start codon were clustered at 70% identity, and one member of each cluster was selected to generate a sequence alignment. Multiple sequence alignment (MSA) of the selected members of the IS110 and IS1111 family members, along with the Piv inversion protein of *Moraxella lacunata* comprising a total of 197 sequences was performed in MEGA (Molecular Evolutionary Genetics Analysis version 11[39]) using default settings of Clustal Omega[40]. A phylogenetic tree was generated in MEGA using the default settings for a neighbour-joining tree and rooted using the midpoint. The MSAs were further analysed with the WebLogo tool[41] to identify conserved residues. A per cent identity matrix was created using Clustal Omega. 3D structure prediction of protein sequences was performed using AlphaFold2[29] to generate the monomer, dimer and tetramer structures and visualised on Pymol (Schrodinger, LLC. The PyMOL Molecular Graphics System, Version 1.8. 2015). ISEc11 and ISEc21 were used as representative members of the IS110 and IS1111 families. Protein sequence conservation was analysed using the Consurf tool[42].

### Genome mining of consensus target site of IS110 and IS1111 family members

DNA sequences for 349 total IS110 and IS1111 family members extracted from ISFinder database were searched against the pre-formatted non-redundant nucleotide database (version date 2022-11-21) using BLAST+ version 2.13.0, restricting to bacterial taxonomic IDs. The BlastN output was filtered by E value of 0, 100% identity, and 95% identity; subject length ≥100,000 and ≤10,000,000; query coverage 100%. This resulted in the collection of GenBank coordinates where each IS was inserted. From those coordinates, an extra 200 base pairs were selected from each side of each identified IS using a custom script and a new database of IS ±200 bp on each side was created. The flanking sequences were concatenated to reform the pre-insertion sites. An MSA of all pre-insertion sites for each member of the IS110 and IS1111 family was used to identify and remove identical sequences in order to generate a non-redundant MSA of unique insertion events. To find the consensus target sequence for each IS, the resulting unique flanking sequences were aligned using Clustal Omega and the MSA was analysed using a custom script to generate a WebLogo[41] for 20 bp on each side of the IS.

### Molecular cloning and plasmids

*E. coli* strain DH5α was used for molecular cloning, and all molecular cloning was performed by the Gibson cloning method unless specified otherwise and introduced into their host using electroporation with selection for a relevant antibiotic. Transformants were plated onto selective agar plates and grown overnight at 37 °C. Plasmid DNA was prepared using the ISOLATE II Plasmid Mini Kit (Bioline). The sequence of a selected member of the IS110 (ISEc21) and IS1111 (ISEc11) family was retrieved from ISfinder and used to locate an insertion point for each IS. Spans consisting of the IS and 100 bp from each flank, 4360197 to 4361839 in CP053751.1 for ISEc11 and 1801957 to 1803530 in LT903847.1 for ISEc21, were synthesised as a gBlock (Integrated DNA Technologies; IDT) that was cloned into the BamHI site in pUC19, under the control of *lac* promoter to generate pSFA04 for ISEc11 and pSFA09 for ISEc21 (Supplementary Data 1). A derivative containing the T7 promoter introduced at the upstream BamH1 site was also constructed. The DNA segments containing the respective derived target sequences, 47 bp from the left side of the IS and 33 bp from the right derived for ISEc11 and 55 bp from the left side of the IS and 51 bp from the right for ISEc21, were also synthesised as gBlocks and cloned into pRSF Duet-1. pUC19-based plasmids with 100 bp flanking the IS on each side were also constructed by PCR amplification of the appropriate region of published plasmids containing ISKpn4[2], 2547772 to 2549449 of CP034908.2 for ISPa11 and 1889 to 3601 of EF648212.1 for ISPst6. An upstream T7 promoter was then introduced. All other regions, *tnpEc11*, *tnpEc21*, NCRs, the *catA1* chloramphenicol resistance gene with its endogenous upstream promoter, and the *mCherry* gene, etc., were synthesised as gBlocks and introduced into appropriate locations in pUC19, pRSF Duet-1 or pCDF Duet-1 or their previously constructed derivatives. All plasmids used with a brief description and links to sequences are listed in Supplementary Data 1. The sequence of the NCRs from ISs used in this study is in Supplementary Table 1.

### Purification and analysis of Tnp RNP complexes

BL21 (DE3) *E. coli* cells containing a pCDF-Duet-1 plasmid designed to produce a transposase with an N-terminal 6xHis-Maltose-Binding-Protein (MBP), a TEV cleavage site and a C-terminal Strep-TagII either alone or with a pUC19-based plasmid carrying the corresponding IS with an upstream T7 promoter were grown in 500 mL of LB supplemented with spectinomycin (100 μg/mL) and ampicillin (100 μg/mL) until OD$_{600nm}$ of 0.6–0.8 when 0.5 mM IPTG (isopropyl β-D-1-thiogalactopyranoside) was added to induce the expression of the tagged transposase and transcription of the IS and grown for an additional 16 h at 18 °C. Cells were pelleted by centrifugation at 5000*g* for 10 min, resuspended in Lysis buffer (100 mM Tris-HCl pH 8.0, 500 mM NaCl, 5% Glycerol, 1 mM TCEP) and lysed by sonication. Cell debris was removed by centrifugation (30 min at 17,000*g*), and the clarified supernatant was loaded onto 1 mL of StrepTactin-XT (high capacity) affinity chromatography column (IBA Lifesciences) and washed with 5 column volumes of Lysis buffer. The protein was eluted with Lysis buffer containing 50 mM Biotin, and the fractions containing the transposase were combined and concentrated using 30 kDa Amicon centrifugal concentrators (Merck) and stored at −80 °C.

Samples were loaded on NuPAGE™ 4–12% Bis–Tris Protein Gels (Invitrogen), and electrophoresis was performed at 200 V for 20 min in MES running buffer (Invitrogen). The gels were stained with Coomassie Brilliant Blue and destained in water. To analyse nucleic acids bound to the transposase, 10 μL samples were incubated with either 1 μL of DNase I (18 U/μL) or 1 μL of RNase (10 U/μL) at 37 °C for 30 min. Samples were then mixed with 2× RNA loading dye (95% formamide, 10 mM EDTA, bromophenol blue, xylene cyanol), heated at 98 °C for 2 min to denature before loading on a 7% (19:1) TBE–Urea (7 M) denaturing polyacrylamide gel and electrophoresis was performed at 200 V for 30 min using 1× TBE running buffer. The gels were stained with SYBR™ Gold (Invitrogen) for nucleic acid visualisation.

## Small RNA-Sequencing and analysis

In total, 100 μL of the purified transposase alone and Tnp-RNA (RNP) complex were incubated with 5 μL of DNase (20 U/μL) for 30 min at 37 °C, then 5 μL of Proteinase K was added and incubated for 30 min at 37 °C. The RNA was ethanol precipitated, and the pellet was washed with 70% ethanol and resuspended in 15 μL MilliQ water. 2 μg of purified RNA was used to prepare RNA libraries for sequencing. Colibri Stranded RNA Library Prep Kit for Illumina Systems (Thermo Fisher Scientific) was used, and the libraries were prepared using the manufacturer's protocol and sequenced using an iSeq100 System (Illumina). These reads were then aligned to the sequence of their corresponding IS using the Burrows–Wheeler Aligner alignment tool. The coverage plot was visualised using Integrative Genomics Viewer (IGV)[43] or Tablet v1.21.02.08[44]. RNA fold prediction was performed with MXfold2[30].

## Mini-circle formation and in vivo transposition assays

Mini-circle formation experiments were conducted using plasmid DNA from *E. coli* strain DH5α containing the pUC19 plasmid containing the full-length IS sequences with 100 bp flanking sequences isolated using a standard plasmid isolation kit (Bioline). PCR primers faced outwards from within the IS. The primer set used for each IS is listed in Supplementary Table 2. PCR was performed in a final reaction volume of 20 μL using Phusion™ reaction buffer, 0.2 mM of dNTPs, 0.5 μM of each primer, and 1 U of Phusion™ polymerase (New England Biolabs) and 1 μl of template DNA. Cycling conditions were a denaturation cycle (98 °C, 2 min) followed by 35 cycles of denaturation (98 °C, 15 s), annealing (52 °C, 20 s), extension (72 °C, 20 s), and a final cycle of amplification (72 °C, 40 s). The PCR product was run on a 1% agarose gel with a 1 kbp Plus DNA ladder (New England Biolabs). Amplicons were isolated using a PCR gel extraction kit (Bioline) and Sanger sequenced to identify the minicircle junction product.

In vivo transposition assays were carried out in *E. coli* DH5α or BL21(DE3) containing various combinations of plasmids including a pDonor plasmid carrying the IS or derivatives with smaller or other regions flanked by IS ends and flanking target (in pUC19; ampicillin resistance) and a pTarget plasmid containing the target in pRSF Duet-1 (kanamycin resistance) and, if required, a third plasmid carrying accessory factors, e.g. a pSeekRNA in pCDF Duet (streptomycin resistance). Isolated plasmid DNA was diluted to 10 ng/μL, and 10 ng was used as a template for PCR to detect transposition. Primer pair sets were complementary to the IS (R) in the *tnp* gene and the pTarget plasmid backbone (F) and the alternate pair (F) on IS and (R) on pTarget plasmid backbone (Supplementary Table 3). The PCR reaction was performed as for mini-circle formation, with the following changes, extension (72 °C, 30 s), and final cycle of amplification (72 °C, 60 s). The PCR products were isolated and sequenced for minicircle formation.

## mCherry reporter transposition assay in trans

A pDonor plasmid containing the *tnp* and seekRNA followed by the HDV sequence was cloned under the control of a T7 promoter, while a strong T7 terminator sequence was cloned in the 3′ of the HDV sequence to prevent downstream gene expression. The *mCherry* cargo gene is flanked by L and R sequences of the corresponding IS, as described above. The length and sequence of each of these components are listed in the supplementary material (Supplementary Data 1). The pTarget plasmid contains the IS target sequence cloned downstream of a T7 promoter to detect transposition via mCherry expression.

In total, 100 ng of each plasmid (pDonor and pTarget) were either individually transformed or co-transformed into *E. coli* BL21 (DE3) cells via electroporation. Cells were cultured on agar plates containing spectinomycin and kanamycin overnight at 37 °C. Subsequently, a fresh 2 mL LB medium was inoculated with the initial culture and grown until reaching an $OD_{600}$ of 0.6–0.8, followed by the addition of 0.5 mM IPTG and incubation for 4 h at 25 °C.

Cell assays involved transferring 100 μL of each culture into a 96-well Corning clear bottom plate. Cell density was quantified via absorbance at 600 nm, while mCherry fluorescence was measured using a TECAN infinite M1000Pro plate reader in bottom reading mode, with excitation at 587 nm and emission at 610 nm, each with a bandwidth of 5 nm, and the optimal gain set to 100%. To adjust fluorescence measurements for cell density, fluorescence per cell (FOD) was calculated as the fluorescence intensity divided by the $OD_{600}$. The per cent increase in mCherry fluorescence attributable to transposition was determined by dividing the FOD for cells containing pDonor and pTarget by the FOD for pDonor alone. These values were then normalised to the mCherry cargo-only samples and represented a percentage increase in mCherry fluorescence. This analysis was validated through at least three independent transformations and co-transformations as biological triplicates, and Sanger sequencing was performed to validate the transposition of the mCherry cargo gene.

## Quantification of mCherry reporter transposition

Transposition frequency was measured by Flow Cytometry using BD LSRFortessa™ X-20 Flow Cytometer. 100 ng of each plasmid (pDonor and pTarget) (Supplementary Data 1) was co-transformed into *E. coli* BL21(DE3) cells via electroporation. Cells were plated on fresh agar plates containing spectinomycin, kanamycin and 0.1 mM IPTG to induce expression of the transposase and transcription of seekRNA, as well as expression of mCherry after transposition. The plates were incubated for 16 hours at 37 °C, followed by 4 h at room temperature. Entire agar plates containing hundreds of colonies were scraped and evenly resuspended in 1 mL of LB media. Cells were diluted 1 in 2 in PBS (Phosphate Buffered Saline, pH 7.4) and run on the Flow Cytometer. Around 35,000–70,000 cells were run for each sample until at least 25,000 events were recorded, which were gated for single live cells. The cells were counted using forward, and side scatter channels, and mCherry fluorescence intensity was detected per cell. The transposition frequency was measured according to the number of single cells with high levels of mCherry fluorescence intensity over a total number of single cells. Each set of samples was created by three independent transformations as biological repeats, and the transposition frequency was plotted as bar graphs. Sanger sequencing was performed to validate the transposition of the mCherry cargo gene. Flow cytometry data was analysed using FlowJo™ v10.10 Software (BD Life Sciences).

## Reagents

Commercially available reagents with catalogue numbers are described in Supplementary Data 2.

## Reporting summary

Further information on research design is available in the Nature Portfolio Reporting Summary linked to this article.

# Data availability

The data supporting the findings of this study are available from the corresponding authors upon request. Genbank genomes and annotated IS from ISFinder used to generate the GitHub are publicly available. Small RNA-Seq data from Illumina sequencing datasets generated in this study are available on NCBI Sequence Read Archive under PRJNA1091059. Source data for the figures and Supplementary Figs. are provided as a Source Data file. Source Data includes a txt file for the fasta format of curated transposase sequences for Fig. 1 and Supplementary Figs. 1 and 2. Source data are provided in this paper.

# Code availability

The code generated for Genbank genomes and annotated IS from ISFinder are publicly available, and target sequence consensus creation is available from https://github.com/AtaideLab/Targets[31].

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

## Acknowledgements

We thank the curators of the ISFinder database for continuing to update this valuable resource. This research was supported by the Sydney Informatics Hub and Sydney Analytical, Core Research Facilities of the University of Sydney. We especially thank Cali Willet for the help with the pipeline. This work was support by the University of Sydney and DVCR Strategic Research Impact Fund POC 16-2023 to S.F.A. R.M.H. was supported by NHMRC Investigator grant GNT1194978.

## Author contributions

R.M.H. and S.F.A. conceived the study. S.F.A., R.S., C.H.P. and R.M.H. designed experiments, and S.F.A. and R.S. designed the computational strategy. S.F.A., R.S. and R.M.H. performed computational analysis and S.F.A., R.S. and C.H.P. performed experiments. S.F.A. and R.M.H. supervised the research. S.F.A., R.S. and R.M.H. analysed and interpreted the data. R.S. and S.F.A illustrated the paper with input from R.M.H. R.M.H. and S.F.A. wrote the paper. All authors reviewed the final paper.

## Competing interests

R.S., R.M.H. and S.F.A. are inventors of patents related to this work. The remaining authors declare no competing interests.
