## [Peer Review File · Nature Communications]

A programmable seekRNA guides target selection by IS1111 and IS110 type insertion sequencesEditorial Note: This manuscript has been previously reviewed at another journal that is not operating a transparent peer review scheme. This document only contains reviewer comments and rebuttal letters for versions considered at Nature Communications.

REVIEWER COMMENTS

Reviewer #1 (Remarks to the Author):

In the revised manuscript authors addressed most of my comments or provided reasonable explanations.

Reviewer #3 (Remarks to the Author):

In my opinion, this manuscript contains some important information concerning the transposition mechanism of the IS110 and IS1111 family of IS. The results nicely complement those from the Hsu lab (Durrant et al DOI: 10.1101/2024.01.24.577089).

The authors use the IS110/IS1111 collection contained in ISfinder as a base for their analysis. The analysis opens with a comprehensive analysis of the transposase (although a colleague brought to my attention that the phylogenetic tree and transposase alignments have been available for some time at https://tncentral.ncc.unesp.br/TnPedia/index.php/IS_Families/IS110_family). The authors also include a comparison of transposase secondary structure obtained using Alphafold.

The authors provide several important lines of evidence which distinguish the IS110 and IS1111 groups. These include a transposase-based tree clearly distinguishing the two groups and transposase alignments indicating that the N-terminal catalytic domain of both groups is conserved while the c-terminal domain of IS110 group members is more variable, both of which were previously available; and the presence of sub terminal inverted repeats in IS1111 group members, previously identified by the Hall laboratory.

However, the novel characteristic identified in the present manuscript is that while both IS110 and IS1111 group members include long non-coding regions (NCR), in the IS110 group it is located upstream of the transposase gene while for the IS1111 group, it is located downstream.

Together, these differences now provide a convincing argument for the separation of the IS110 family into two well defined groups.

In the case of the IS1111 group member, ISEc11, the authors establish, using a plasmid-based assay, that the (downstream) NCR region is strictly required for formation of a circular ISEc11 transposition intermediate (as judged by PCR) and that an inactivated IS carrying a deletion of this region can be complemented either in trans or in cis. The IS uses a specific target site and insertion occurred at the correct position and orientation. Moreover, deletion of the target DNA sequences which flank the IS also prevented circle formation.

It is then shown that the NCR region encodes an RNA which co-purified with the transposase. Its sequence mapped to the 3'NCR. They observe similar RNS species for two additional IS1111 members, ISKpn4 and ISPa11 which exhibit different target sequences.

Intriguingly, they note that each of the folded RNA species includes short sequences in a similarly placed loop region, one of which is complementary to the top and the other to the bottom strand of the target DNA. They clearly demonstrate that changing these sequences permits insertion into target sites with the appropriate complementary sequence changes.

An important piece of information is provided concerning the difference between the IS1111 group and the IS110 group. ISEc21 (IS110) includes an NCR upstream of the transposase gene. Its deletion prevents circle formation which can be restored by providing NCR in trans or in cis. Again the target sequence on each side of the IS is shown to be required for transposition and circle formation and an RNA corresponding in sequence to the upstream NCR co-purified with the transposase.

Importantly, in this RNA there is also a small complementary sequence similarly placed as in the IS1111 RNAs. However, where in the IS1111 RNAs the top strand complementarity is located 5' to the bottom strand match, this is reversed for ISEc21.

L:118. it unnecessary to include the alignments in Supplementary figure 1? These have been published previously and are available online. And the importance of the SG residues had been shown. I am not sure why this is included here.

L: 127. A similar phylogenetic tree showing separation of IS110 and IS1111 groups is available online.

L:174. Does the absence of the uninterrupted target sequence imply a replicative copy-out-paste-in mechanism? Have the authors considered a genetic test as was used in the case of IS492 to show reconstitution of the target site? Could this be a difference in IS110 (IS492) and IS1111 (ISEc11)?

L:204. Some comment concerning RNA length is required here. Are the sequences shown in Fig. 3g,h,I the full extent of the NCR species of simply specifically chosen sections?

L: 213. The authors must make clear what they mean by longer and shorter RNA and define them in the text. Also, they use the term seekRNA prematurely without any other information. This implies that the RNA function is to "seek" the target which presupposes a function. Its function is presumably to bridge the circular IS and its target?

L: 233. This is not strictly true. Reprogramming means that one is converted into the other. This is not demonstrated. The two are simply similar with slightly different readout sequences. This paragraph should either be rephrased or removed.

L: 241 and on. This is indeed reprogramming. But we suddenly come across a 154nt NC RNA. The authors should define these different length species in the text (L: 213). This brings up the question of how they are produced – different promoters or transposase-mediated processing as occurs with guide RNAs.

L:253. Again, the short NC RNA is now used. It is shown to have enhanced activity. How do the authors explain this?

L:257. I am left wondering why the authors think that this is necessarily to include. The "cargo" is really short and surely the argument is quite obvious. The relevant question here would probably be how much cargo does this system permit to carry. I suggest that the section is removed.

L:282. The authors clearly state here that they isolated a long and short RNA form as they should in the case of the IS1111 examples.

L:297: Again, this is not Natural Reprogramming.

L:312: Number of copies of the IS or number of copies of the potential sites? The 7000 observed in *B.pertussis* is extraordinary since, according to the Parkhill lab, the genome also carries hundreds of copies of IS481 as well.

L:321. Some or all of the IS ends in IS finder? The way that this is phrased seems to suggest that all

IS110/IS1111 members have miss-annotated ends. Please be more precise. Give a percentage perhaps.

L:399. Is always dangerous to use "for the first time" In this case, the manuscript from the Hsu lab also shows this. Please delete.

L:348. It seems possible that the long RNA is produced from a given promoter and that the shorter version is generated by processing of the longer version as with some of the guide RNAs.

The RNA described here shows complementarity to the target. Since the authors call it a seekRNA, I am intrigued how the authors imagine the RNA "seeks" the target. I note that in the Durrant et al preprint, the authors show an RNA with two loops: one which, like those shown here, would recognize the target and the other which they propose to recognize the IS circle or donor junction. Do the authors find such a second loop in their longer RNAs (for example)? In this light

Reviewer #3 (Remarks on code availability):

I am not competent to assess the code

Reviewer comments

Reviewer #1 (Remarks to the Author):

In the revised manuscript authors addressed most of my comments or provided reasonable explanations.

** no changes requested.

Reviewer #3 (Remarks to the Author):

In my opinion, this manuscript contains some important information concerning the transposition mechanism of the IS110 and IS1111 family of IS. The results nicely complement those from the Hsu lab (Durrant et al DOI: 10.1101/2024.01.24.577089).

The authors use the IS110/IS1111 collection contained in ISfinder as a base for their analysis. The analysis opens with a comprehensive analysis of the transposase (although a colleague brought to my attention that the phylogenetic tree and transposase alignments have been available for some time at https://tncentral.ncc.unesp.br/TnPedia/index.php/IS_Families/IS110_family). The authors also include a comparison of transposase secondary structure obtained using Alphafold.

The authors provide several important lines of evidence which distinguish the IS110 and IS1111 groups. These include a transposase-based tree clearly distinguishing the two groups and transposase alignments indicating that the N-terminal catalytic domain of both groups is conserved while the c-terminal domain of IS110 group members is more variable, both of which were previously available; and the presence of sub terminal inverted repeats in IS1111 group members, previously identified by the Hall laboratory.

However, the novel characteristic identified in the present manuscript is that while both IS110 and IS1111 group members include long non-coding regions (NCR), in the IS110 group it is located upstream of the transposase gene while for the IS1111 group, it is located downstream.

Together, these differences now provide a convincing argument for the separation of the IS110 family into two well defined groups.

In the case of the IS1111 group member, ISEc11, the authors establish, using a plasmid-based assay, that the (downstream) NCR region is strictly required for formation of a circular ISEc11 transposition intermediate (as judged by PCR) and that an inactivated IS carrying a deletion of this region can be complemented either in trans or in cis. The IS uses a specific target site and insertion occurred at the correct position and orientation. Moreover, deletion of the target DNA sequences which flank the IS also prevented circle formation.

It is then shown that the NCR region encodes an RNA which co-purified with the transposase. Its sequence mapped to the 3'NCR. They observe similar RNS species for two additional IS1111 members, ISKpn4 and ISPa11 which exhibit different target sequences.

Intriguingly, they note that each of the folded RNA species includes short sequences in a similarly placed loop region, one of which is complementary to the top and the other to the bottom strand of the target DNA. They clearly demonstrate that changing these sequences

permits insertion into target sites with the appropriate complementary sequence changes.

An important piece of information is provided concerning the difference between the IS1111 group and the IS110 group. ISEc21 (IS110) includes an NCR upstream of the transposase gene. Its deletion prevents circle formation which can be restored by providing NCR in trans or in cis. Again the target sequence on each side of the IS is shown to be required for transposition and circle formation and an RNA corresponding in sequence to the upstream NCR co-purified with the transposase.

Importantly, in this RNA there is also a small complementary sequence similarly placed as in the IS1111 RNAs. However, where in the IS1111 RNAs the top strand complementarity is located 5' to the bottom strand match, this is reversed for ISEc21.

L:118. it unnecessary to include the alignments in Supplementary figure 1? These have been published previously and are available online. And the importance of the SG residues had been shown. I am not sure why this is included here.

** First, this is a Supplementary Figure and its removal would make the manuscript rather difficult to follow in places. For example, the online work does not mention the SG conservation at all but we have located these residues on the AlphaFold models we show in Figures and they are discussed in the text. These residues are not marked on the online alignment, and it looks like the SG may not be conserved in the IS1111 group. A clearer version would be needed to be sure.

** Hence, we have retained this important Figure.

** The reviewer may not have looked at the online version to see that the resolution is such that it's almost impossible to read, requires lots of guessing.

** We know of no recent alignments of the complete IS1111 family separate from the IS110 (and perhaps other families currently in this group) published previously. The online version is also problematic as it can be changed by the site owners at any time or the site may disappear at some time in the future.

L: 127. A similar phylogenetic tree showing separation of IS110 and IS1111 groups is available online.

** Again, this is a Supplementary Fig and the text would be very hard to follow without it. So, we do not want to lose it. For example, this Figure places the IS we use (they are red) and the founders for IS1111 and IS110 families within the context of specific subfamilies in the tree that are discussed in the text. Hence, we have retained this key Figure.

** Furthermore, the two versions are not the same. Our phylogeny was carefully curated, the online version is not. This is an important difference as inclusion of multiple versions (close relatives) of only some items is known to seriously affect the topology.

** See also comments above. The very low resolution of the online version means that we cannot read any of the IS names at any magnification. So, we cannot tell if that tree and ours reveal the same IS subgroups. Hence, it could not work as a substitute for our Fig.

L:174. Does the absence of the uninterrupted target sequence imply a replicative copy-out-paste-in mechanism? Have the authors considered a genetic test as was used in the case of IS492 to show reconstitution of the target site? Could this be a difference in IS110 (IS492) and IS1111 (ISEc11)?

** We have been careful to not speculate on mechanism at this stage. We know too little and there is no precedent. We have deliberately not used terms such as “copy-out -paste-in” as this term is strongly associated with the DDE transposases and the mechanism has features we do not see. Hence, its use risks leading readers to think the mechanisms would be the same. Clearly, they are not. More work will be needed here. It’s possible that the mechanism could involve a rolling circle mechanism. However, a comment has been added about this in the Discussion.

** I assume the “genetic test” referred to relates to the phenotypic consequences of the location of IS492. In addition, reconstitution of the empty IS492 target was at low frequency relative to circle formation. We have no need for this approach, as our data is clear and has been tested on several IS1111 family members and on ISEc21 (IS110 family). No change needed.

** IS492 belongs to the potential third family, currently known as the Piv group. Perhaps this family is a little different. Work will be needed to look at this. That’s way beyond the scope of the current study. No change needed.

L:204. Some comment concerning RNA length is required here. Are the sequences shown in Fig. 3g,h,i the full extent of the NCR species of simply specifically chosen sections?

** Agreed. The information was in the Fig. but is needed in the text. We have added a few sentences to this section to explain our use of the various terms.

** The lengths shown in Fig 3 are experimentally determined not “chosen”. That they represent the actual sizes of the RNA molecules detected is now clearly stated in the text.

L: 213. The authors must make clear what they mean by longer and shorter RNA and define them in the text. Also, they use the term seekRNA prematurely without any other information. This implies that the RNA function is to “seek” the target which presupposes a function. Its function is presumably to bridge the circular IS and its target?

** Yes, to defining long and short in the text and this has been done (see above). We have also modified the relevant sentence in the last paragraph of the Introduction.

** The seekRNA term does not seem to us to be used prematurely. It’s first used after the target matching role, which explains the term, has been found (except for the section heading). So, no change made.

** Re the last sentence of this comment: Our data does not support any role other than target recognition (seeking the target). The reviewer may be referring to the “bridging” role proposed in the preprint cited by the reviewer. However, we have shown that the segment that they predict recognises the IS ends in the single IS (it’s a Piv group IS) for which they present data is not in our fully functional short seekRNA. So, definitely no “bridge” for the ISs we are studying here.

L: 233. This is not strictly true. Reprogramming means that one is converted into the other. This is not demonstrated. The two are simply similar with slightly different readout sequences. This paragraph should either be rephrased or removed.

** We don’t understand this comment. Perhaps semantics. The OED definition is: “**to program anew**. especially : to revise or write a new program for. reprogram a computer. intransitive verb. : to rewrite or revise a program especially of a computer.”

** Are targets that share only 3 of 13 bases really “slightly different readouts”?

** In the case described, the stems of the seekRNAs are almost identical but the heads including the target-specific residues have been replaced. The target-specific residues match the observed targets, so the target has been changed.

** We have expanded and slightly changed the wording of this paragraph.

L: 241 and on. This is indeed reprogramming. But we suddenly come across a 154nt NC RNA. The authors should define these different length species in the text (L: 213). This brings up the question of how they are produced – different promoters or transposase-mediated processing as occurs with guide RNAs.

** The 154 nt RNA is the long seekRNA as now defined in the text (see above). As these are in vivo experiments, it may well be processed to the seekRNA (the shorter length observed in earlier sections).

** As there is only one promoter, the long seekRNA would be processed, presumably by the transposase, to generate the seekRNA.

L:253. Again, the short NC RNA is now used. It is shown to have enhanced activity. How do the authors explain this?

** We do not know. Perhaps, because it is the functional seekRNA and the untrimmed regions in the longer form interfere.

L:257. I am left wondering why the authors think that this is necessarily to include. The “cargo” is really short and surely the argument is quite obvious. The relevant question here would probably be how much cargo does this system permit to carry. I suggest that the section is removed.

** This is a small section, and its removal would not significantly enhance the manuscript. So, no change.

** It's useful for future studies to show that the system does work in trans and that insertions can be made close to both ends of the IS.

L:282. The authors clearly state here that they isolated a long and short RNA form as they should in the case of the IS1111 examples.

** No change requested.

L:297: Again, this is not Natural Reprogramming.

**See above.

L:312: Number of copies of the IS or number of copies of the potential sites? The 7000 observed in *B.pertussis* is extraordinary since, according to the Parkhill lab, the genome also carries hundreds of copies of IS481 as well.

** As clearly stated in the text and in Fig. 8, it is the number of total locations found in all of the genomes examined. We have added some words to try to clarify this. We don't know the relevance of IS481.

L:321. Some or all of the IS ends in IS finder? The way that this is phrased seems to suggest

that all IS110/IS1111 members have miss-annotated ends. Please be more precise. Give a percentage perhaps.

** The sentence ends “in some cases”. So, it never said or implied “all”. “In some cases” has been moved forward in the sentence to overcome any potential for confusion.

** We have also added some information on how many did not yield logos – it’s a majority for IS110 family members but also a lot for the IS1111s.

** We believe the IS110 family problem lies with the difficulty in finding the correct ends especially granted target conservation – that will need experimental determination of the sequence of the junction in the circular intermediate and now have said so.

L:399. Is always dangerous to use “for the first time” In this case, the manuscript from the Hsu lab also shows this. Please delete.

** First, there is a preprint not a manuscript. Moreover, the single IS they have used is from the probable Piv family not the IS1111 family where our work is the first. The sentence has been expanded to be clearer and separate the IS1111s from the IS110.

L:348. It seems possible that the long RNA is produced from a given promoter and that the shorter version is generated by processing of the longer version as with some of the guide RNAs.

** There’s only 1 promoter, the one generated via formation of the circular form. So, yes the longer RNA is made and most would have been processed as it’s found only in very small amounts associated with the transposase.

** A sentence on this has been added at the end of the first paragraph in the section headed “IS1111 family transposases co-purify with NCR-derived RNA” and also in the Discussion.

The RNA described here shows complementarity to the target. Since the authors call it a seekRNA, I am intrigued how the authors imagine the RNA “seeks” the target. I note that in the Durrant et al preprint, the authors show an RNA with two loops: one which, like those shown here, would recognize the target and the other which they propose to recognize the IS circle or donor junction. Do the authors find such a second loop in their longer RNAs (for example)? In this light

** No, there is no such second loop in the seekRNA (the short NCR-derived RNA we found) which was shown to be fully functional and reprogrammable. The preprint mentioned seems to make assumptions about the length of the RNA and uses the entire NCR. It could be that for the Piv type IS their assumption is correct. Only experimentation will clarify this.

REVIEWERS' COMMENTS

Reviewer #3 (Remarks to the Author):

OK. No more comments